# Voluntary Running Improves Behavioral and Structural Abnormalities in a Mouse Model of CDKL5 Deficiency Disorder

**DOI:** 10.3390/biom13091396

**Published:** 2023-09-15

**Authors:** Nicola Mottolese, Beatrice Uguagliati, Marianna Tassinari, Camilla Bruna Cerchier, Manuela Loi, Giulia Candini, Roberto Rimondini, Giorgio Medici, Stefania Trazzi, Elisabetta Ciani

**Affiliations:** 1Department of Biomedical and Neuromotor Sciences, University of Bologna, 40126 Bologna, Italy; 2Department of Medical and Surgical Sciences, University of Bologna, 40126 Bologna, Italy

**Keywords:** *Cdkl5* KO mice, voluntary exercise, brain development, dendritic pathology, neuronal survival, neuroinflammation

## Abstract

Cyclin-dependent kinase-like 5 (*CDKL5*) deficiency disorder (CDD) is a rare neurodevelopmental disease caused by mutations in the X-linked *CDKL5* gene. CDD is characterized by a broad spectrum of clinical manifestations, including early-onset refractory epileptic seizures, intellectual disability, hypotonia, visual disturbances, and autism-like features. The *Cdkl5* knockout (KO) mouse recapitulates several features of CDD, including autistic-like behavior, impaired learning and memory, and motor stereotypies. These behavioral alterations are accompanied by diminished neuronal maturation and survival, reduced dendritic branching and spine maturation, and marked microglia activation. There is currently no cure or effective treatment to ameliorate the symptoms of the disease. Aerobic exercise is known to exert multiple beneficial effects in the brain, not only by increasing neurogenesis, but also by improving motor and cognitive tasks. To date, no studies have analyzed the effect of physical exercise on the phenotype of a CDD mouse model. In view of the positive effects of voluntary running on the brain of mouse models of various human neurodevelopmental disorders, we sought to determine whether voluntary daily running, sustained over a month, could improve brain development and behavioral defects in *Cdkl5* KO mice. Our study showed that long-term voluntary running improved the hyperlocomotion and impulsivity behaviors and memory performance of *Cdkl5* KO mice. This is correlated with increased hippocampal neurogenesis, neuronal survival, spine maturation, and inhibition of microglia activation. These behavioral and structural improvements were associated with increased BDNF levels. Given the positive effects of BDNF on brain development and function, the present findings support the positive benefits of exercise as an adjuvant therapy for CDD.

## 1. Introduction

Cyclin-dependent kinase-like 5 (*CDKL5*) deficiency disorder (CDD) is a rare neurodevelopmental disease caused by mutations in the X-linked *CDKL5* gene, a major role of which is to encode a serine-threonine kinase that is highly expressed in the brain [1,2,3]. CDD is a severe condition that is characterized by infantile-onset refractory epilepsy, hypotonia, developmental delay, intellectual disability, and visual impairment [4,5]. There is currently no targeted therapy for CDD that is able to address the underlying problems of the disorder.

To gain a deep understanding of the effects of *CDKL5* deficiency in brain development, and to investigate possible therapeutic approaches for CDD, knockout (KO) mouse models for *Cdkl5* were developed. The *Cdkl5* KO mouse exhibits numerous behavioral deficits that are reminiscent of human symptomatology across motor, sensory, cognitive, and socio-emotional domains, including severe learning and memory impairment, autistic-like behaviors, and motor stereotypies [6,7,8,9]. The behavioral deficits are associated with neuroanatomical alterations, detected in the cortex and hippocampal region: reduced dendritic branching and spine maturation [7,8,10,11,12], defects in connectivity [8,9,13], and an increased status of microglia activation, highlighted by a change in microglial cell morphology and number, increased levels of AIF-1 and pro-inflammatory cytokines, and activation of STAT3 signaling, typical markers of an M1 pro-inflammatory phenotype [14,15]. In addition, the hippocampus of *Cdkl5* KO mice was found to be characterized by decreased survival of newborn cells in the dentate gyrus and of CA1 pyramidal neurons [12,16,17], suggesting that loss of Cdkl5 has an endangering action that is likely to promote neuronal death.

Physical activity is generally believed to enhance learning and memory and delay cognitive decline associated with aging in humans [18,19,20,21,22]. Studies in rodents have also demonstrated that voluntary running improves performance in hippocampus-dependent learning tasks [23,24,25], reduces anxiety-like behaviors [26], and ameliorates motor skills [27]. Running induces widespread structural alterations in the hippocampus and cortex. It is known to increase cell proliferation and neurogenesis in the hippocampus of both rodents and humans [28,29,30,31]. Furthermore, physical exercise increases spine density and dendritic spine length in the cortex and hippocampus of rodents [32,33], and has been reported to promote mature spine formation [34,35]. It has a positive effect on neuroinflammation, attenuating the age-dependent increase in microglia activation [36,37,38], as measured by the staining intensity of CD11b, in a transgenic mouse model of Alzheimer’s disease, or LPS-induced IL-1β production in aged rats, indicating that exercise robustly reduces the potentiated inflammatory response of aged hippocampal microglia to pro-inflammatory stimulation.

Exercise also significantly alters the microenvironment of the hippocampus in both rodents and humans, with increased growth factor expression [39,40,41]. In particular, it has been extensively demonstrated that BDNF levels increase after exercise in both rodents [41,42,43] and humans [44,45,46]. Interestingly, an exercise-induced increase in cognitive performance correlates with BDNF concentrations [43,47].

Various studies have been conducted with the objective of developing a therapy aimed at the biological and genetic basis of CDD [8,15,48,49,50,51,52,53,54,55]. Some of these therapeutic approaches tested on *Cdkl5* KO mice have been shown to be effective in ameliorating neuroanatomical and behavioral defects. Among these, we recently showed that treatments that act on brain-derived neurotrophic factor (BDNF) signaling, either by mimicking the BDNF function, boosting the BDNF/TrkB pathway, or increasing BDNF expression, had positive effects on the neurological phenotypes of *Cdkl5* KO mice [15,49,55].

In view of the positive effects of voluntary running on the brain of mouse models of various human neurodevelopmental disorders [26,56,57,58], which are accompanied by an increase in BDNF levels, we sought to determine whether exercise, confined to voluntary daily running sustained over a month, could improve brain development and behavioral defects in *Cdkl5* KO mice.

## 2. Materials and Methods

### 2.1. Colony

The mice used in this work derive from the *Cdkl5* KO strain in the C57BL/6N background developed in [7] and backcrossed in C57BL/6J for three generations. Mice for experiments were produced by crossing *Cdkl5* KO heterozygous females (+/−) with wild-type (+/Y) males and they were genotyped using PCR of genomic DNA as previously described [7]. Littermate controls were used for all experiments. The day of birth was designated as postnatal day zero (P0), and animals with 24 h of age were considered as 1-day-old animals (P1). After weaning, mice were housed 3 to 5 per cage, were kept in air-, temperature-, and light-controlled (standard 12 h light/dark cycle) animal rooms, and were fed ad libitum with standard mouse chow and water. The animals’ health and comfort were controlled by the veterinary service. All the experiments were conducted in accordance with the Italian and European Community law for the use of experimental animals and were approved by the Bologna University Bioethical Committee. All efforts were made to minimize animal suffering and to reduce the number of animals used. The number of animals used for each experimental procedure is specified in the figure legends.

### 2.2. Experimental Procedures

A total of 59 adult (4–5-month-old) mice of both genotypes were assigned to one of the following experimental conditions: sedentary (−/Y, *n* = 17; +/Y, *n* = 22) and running (−/Y, *n* = 12; +/Y, *n* = 8); animals were separated into six independent test cohorts (Appendix A). All mice were treated with four BrdU injections on the first experimental day, and housed individually with free access to a running wheel for 1 month or in a cage without a wheel. During the last 4 days of running, mice were behaviorally tested using one or more of the following tests: Hind-Limb Clasping, Object Exploration, and Passive Avoidance, on the days indicated in Figure 1A. The body weight of the animals was recorded every seven days, starting from the beginning of the experiments until the day of sacrifice. Twenty-four hours after the 30th day of voluntary running, the animals were deeply anesthetized with 2% isoflurane (in pure oxygen) and sacrificed. The brains of some animals from each experimental group were processed for histological and immunohistochemistry analysis.

### 2.3. BrdU Treatment

All mice were intraperitoneally injected with 5-bromo-2′-deoxyuridine (BrdU, 150 mg/kg; Sigma-Aldrich, Saint Louis, MO, USA, No. B5002) four times at 2 h intervals during the first day of running, 30 days before sacrifice.

### 2.4. Running Test

To allow voluntary running at any time, four cages were equipped with a plastic running wheel with a 42.7 cm perimeter mounted with a pin on the cage wall. The system was equipped with an electronic device (counter) to measure the number of wheel revolutions. Individual mice were placed in these cages for 30-day sessions. Every morning, after a 24 h session of free wheel running, an operator annotated the number of rotations performed by each mouse and reset the counter.

### 2.5. Histological and Immunohistochemistry Procedures

After the sacrifice, the animals’ brains were quickly removed and cut along the midline. Right hemispheres were Golgi-stained or quickly frozen and used for Western blot analyses (see description below). Left hemispheres were fixed via immersion in 4% paraformaldehyde (100 mM phosphate buffer, pH 7.4) for 48 h, kept in 15–20% sucrose for an additional 24 h, and then frozen with cold ice. Hemispheres were cut with a freezing microtome (Microm GmbH, Walldorf, Germany, No. 910020) into 30 μm thick coronal sections and processed for immunohistochemistry procedures as described below. One out of every eight sections from the hippocampal formation was used for immunohistochemistry for BrdU, doublecortin (DCX), anti-allograft inflammatory factor 1 (AIF-1), and Brain-Derived Neurotrophic Factor (BDNF), following the protocol published in [12,14]. Nuclei were counterstained with Hoechst 33342 (Sigma-Aldrich, Saint Louis, MO, USA, No. H6024). The primary and secondary antibodies used are listed in Appendix A.

### 2.6. Golgi Staining

Hemispheres were Golgi-stained using an FD Rapid Golgi Stain TM Kit (FD NeuroTechnologies Inc., Columbia, MD, USA, No. PK401) as previously described [59]. Hemispheres were cut with a cryostat (Histo-Line Laboratories, Pantigliate, MI, Italy, MC 4000) into 100 μm thick coronal sections that were mounted onto superfrost slides and air-dried at room temperature. After drying, sections were rinsed with distilled water, stained in the developing solution of the kit, and coverslipped with DPX mounting medium (Sigma-Aldrich, Saint Louis, MO, USA, No. 06522).

### 2.7. Image Acquisition and Measurements

Fluorescence images were taken with an Eclipse TE 2000-S microscope equipped with a DS-Qi2 digital SLR camera (Nikon Instruments Inc., Tokyo, Japan, No. 750549) A light microscope (Leica Mycrosystems, Shinjuku City, Tokyo, Japan, No. 512834) equipped with a motorized stage, focus control system, and a color digital camera (Coolsnap-Pro, Media Cybernetics, Rockville, MD, USA, No. A00M82009) were used for bright field images. Measurements were carried out using the Image Pro Plus software version 4.5 (Media Cybernetics, Silver Spring, MD, USA) by investigators blind to the animal’s genotype.

#### 2.7.1. Dendritic Spine Number and Morphology

In Golgi-stained sections, dendritic spines of the basal dendrites of pyramidal neurons located in layers II-III of the motor cortex were manually counted using a 100× oil immersion objective lens (Leitz microscope and objective with 1.4 NA). For each mouse, 10–12 dendritic segments were analyzed, and spine density was expressed as number of spines per μm. The number of spines that belonged to the two different groups (immature spines: filopodium-like, thin, and stubby-shaped; mature spines: mushroom- and cup-shaped) was counted and expressed as a percentage of the total spine number.

#### 2.7.2. Cell Density

The number of BrdU- and DCX-positive cells was counted in the subgranular and granular layers of the dentate gyrus and expressed as number of cells/100 μm. Granule cell density in the upper granular layer of the dentate gyrus and pyramidal neuron density in the CA1 field were evaluated as number of Hoechst-positive nuclei/volume and expressed as number of cells/mm^3^. The number of AIF-1-positive cells in the hippocampus and somatosensory cortex was counted, and cell density was established as AIF-1-positive cells/mm^3^. For all measures, cells were manually counted using the point tool of the Image Pro Plus software (Media Cybernetics, Silver Spring, MD, USA).

#### 2.7.3. Morphometric Microglial Cell Analysis

Starting from 20× magnification images of AIF-1-stained hippocampal and cortical slices, AIF-1-positive microglial cell body size was manually drawn using the Image Pro Plus measurement function and was expressed in μm^2^.

#### 2.7.4. Intensity-Based Analysis of BDNF Staining

The fluorescence signal intensity of BDNF staining in mossy fiber terminals was quantified in the hippocampal formation, starting from 20× magnification images of BDNF-stained hippocampal slices. The average intensities of the fluorescent signals were quantified by determining the number of positive (bright) pixels within three randomly selected areas (50 pixel × 50 pixel) of mossy fibers from the hilus and CA3. The signal intensity was then normalized by subtracting the background intensity of each image. A total of six hippocampal slices were evaluated for each sample.

### 2.8. Western Blotting

For Western blot analysis, brain cortexes were quickly collected and frozen in dry ice. Tissue samples were homogenized in RIPA buffer and quantified using the Bradford method, as previously described [14]. Equivalent amounts (50 μg) of protein were subjected to electrophoresis on Bolt^TM^ 4–12% Bis-Tris Plus gel (Life Technologies Corporation, Carlsbad, CA, USA, No. 04127) and transferred to a Hybond ECL nitrocellulose membrane (GE Healthcare Bio-Science, Piscataway, NJ, USA, No. 10600003). The membrane was cut above the 25 KDa and the 50 KDa band of the marker. The two lower parts of the membrane were separately incubated with an anti-BDNF and an anti-β-actin antibody, respectively, to avoid membrane stripping and reprobing (Appendix A). The primary and secondary antibodies used are listed in Appendix A. Densitometric analysis of digitized Western blot images was performed using ChemiDoc XRS Imaging Systems and the Image Lab^TM^ Software version 5.2 (Bio-Rad, Hercules, CA, USA). This software creates automatic highlights of any saturated pixels in the Western blot images. Images acquired with exposition times that generated protein signals out of a linear range were not considered for quantification.

### 2.9. Behavioral Testing

The behavioral tests were performed during the last week of the voluntary running program (Figure 1A). Mice were allowed to habituate to the testing room for at least 1 h before the test, and testing was always performed at the same time of day. The sequence of the behavioral tests was arranged to minimize the possibility of one test influencing the subsequent evaluation of the next. Behavioral studies were carried out on sedentary and runner *Cdkl5* KO (−/Y) and wild-type (+/Y) male mice (4–5-month-old).

#### 2.9.1. Hind-Limb Clasping

Animals were suspended by their tail for 2 min, and hind-limb clasping time was assessed independently by two operators from video recordings. A clasping event is defined by the retraction of limbs into the body and toward the midline.

#### 2.9.2. Object Exploration in an Open-Field Arena

The animals were placed in the center of a square arena (50 × 50 cm) containing two identical objects (plastic tubes, too heavy for the animal to displace), each of them placed near one of the four corners of the arena (15 cm from each adjacent wall), and were allowed to freely explore the open field for 10 min. Animals’ behavior was monitored using a video camera placed above the center of the arena. Distinct features of locomotor activity, including total distance traveled, average locomotion velocity, and the time spent in the center, were scored using EthoVision XT software version 15.0 (Noldus Information Technology B.V., Wageningen, The Netherlands). To quantify animals’ behavioral response to the presence of the two objects, the total time spent exploring the objects was also scored. The test chambers were cleaned with 70% ethanol between test subjects.

#### 2.9.3. Passive Avoidance Test

For the passive avoidance task, the equipment consisted of a tilting-floor box (47 × 18 × 26 cm) divided into 2 compartments (lit and dark) by a sliding door, and a control unit that incorporated a shocker (Ugo Basile, Gemonio, VA, Italy, No. 7883). Upon entering the dark compartment, mice received a brief mild foot shock (0.4 mA for 3 s) and were removed from the chamber after a 15 s delay. After a retention period of 24 h, mice were returned to the illuminated compartment, and the latency to re-enter the dark chamber was measured, up to 360 s. Chambers were cleaned with 70% ethanol between tests.

### 2.10. Statistical Analysis

Statistical analysis was performed using GraphPad Prism (version 7). Values are expressed as means ± standard error (SEM). All datasets were analyzed using the ROUT method (Q = 1%) for the identification of significant outliers and the Shapiro–Wilk test for normality testing. The significance of results was obtained using Student’s *t*-test and two-way ANOVA or two-way repeated measurement (RM) ANOVA followed by Fisher’s LSD post hoc test, as specified in the figure legends. A probability level of *p* < 0.05 was considered to be statistically significant. The confidence level was taken as 95%. A descriptive statistic of genotype, treatment, and post hoc comparisons is given in Appendix A.

## 3. Results

### 3.1. Running Test and Effect of Voluntary Wheel Running on Body Weight

Adult (4–5-month-old) *Cdkl5* −/Y and wild-type (+/Y) mice were housed individually with free access to a running wheel for 1 month (Figure 1A; mice from here on referred to as “runners”) or in a cage without a wheel (“sedentary”). During the last 4 days of running, mice were behaviorally tested before sacrifice. All mice were treated with 4 BrdU injections on the first running day (Figure 1A).

**Figure 1 biomolecules-13-01396-f001:**
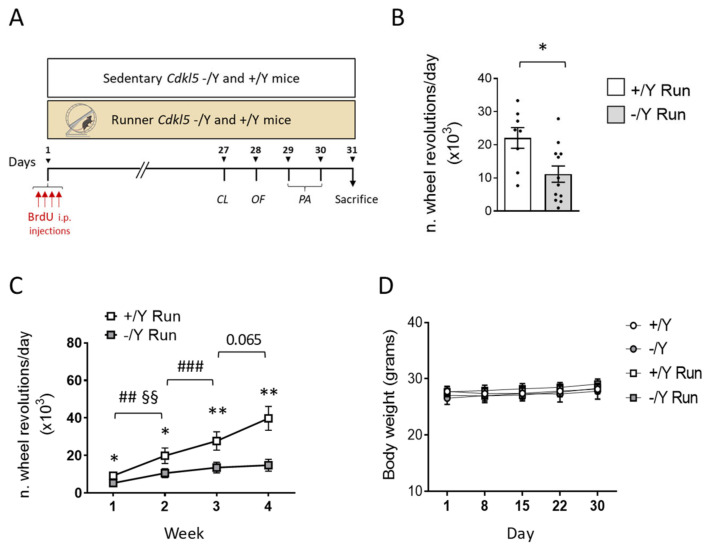
Voluntary wheel running performance in *Cdkl5* −/Y mice. (**A**) Experimental protocol. (**B**) Mean number of revolutions a day performed by 4–5-month-old *Cdkl5* +/Y (*n* = 8) and *Cdkl5* −/Y (*n* = 12) male mice during a 1-month period of voluntary wheel running. The results are presented as means ± SEM. * *p* < 0.05 (two-tailed Student’s *t*-test). (**C**) Mean number of revolutions a day of voluntary wheel running activity performed every week over a one-month period of physical exercise by *Cdkl5* +/Y (*n* = 8) and *Cdkl5* −/Y (*n* = 12) mice. Values are represented as means ± SEM. * *p* < 0.05, ** *p* < 0.01 as compared to the wild-type condition in the same week of physical activity; ## *p* < 0.01, ### *p* < 0.001 in the comparison between *Cdkl5* +/Y mice during different weeks of physical activity; ^§§^
*p* < 0.01 in the comparison between *Cdkl5* −/Y mice during different weeks of physical activity. Fisher’s LSD test after two-way RM ANOVA. (**D**) The graph shows the body weight over a one-month period in sedentary *Cdkl5* +/Y (*n* = 15) and *Cdkl5* −/Y (*n* = 11) mice and in runner *Cdkl5* +/Y (*n* = 8) and *Cdkl5* −/Y (*n* = 12) mice. Values are represented as means ± SEM. Fisher’s LSD test after two-way RM ANOVA. Abbreviations: CL = clasping, OF = open field, PA = passive avoidance.

Wild-type (+/Y) mice were vigorous runners, as they performed an average of 24,108 ± 2702 revolutions daily over a month, whereas *Cdkl5* −/Y mice executed only 11,166 ± 3219 revolutions per day (Figure 1B; *p* = 0.0128). By multiplying 42.7 cm (the length of the perimeter of the running wheel) by the number of revolutions made by the mice in a day, we calculated that the mean daily distances covered by these animals equaled ~10.3 and ~4.7 km for wild-type and *Cdkl5* −/Y mice, respectively. The distance run daily by wild-type and *Cdkl5* −/Y mice was significantly different starting from the first week of running (Figure 1C), a difference that increased in the following weeks, reaching ~63% at the end of the month.

Although running involves high energy expenditure, neither wild-type nor *Cdkl5* −/Y runners showed a body weight reduction over time in comparison with their sedentary counterparts (Figure 1D), suggesting a higher food intake compared with their sedentary counterparts.

### 3.2. Effect of Voluntary Wheel Running on Hyperlocomotion, Impulsivity, and Cognitive Behavior in Cdkl5 KO Mice

Recent studies demonstrated that mice lacking Cdkl5 exhibit hyperlocomotion and impulsivity, resembling core symptoms of attention-deficit hyperactivity disorder (ADHD) [60,61,62,63]. We found that voluntary running restored abnormal elevated locomotor activity (longer distance traveled, Figure 2A, with a higher average speed, Figure 2B) during the open-field exploration period in *Cdkl5* −/Y mice.

Interestingly, the impulsive exploratory behaviors toward the objects inside the open-field arena that were present in sedentary *Cdkl5* −/Y mice were normalized in *Cdkl5* −/Y runners (Figure 2C). To determine whether this increased exploratory behavior was due to a hyperlocomotion in the center of the arena, we compared the time spent in the border area near the walls and in the center of the arena among groups. Wild-type and *Cdkl5* −/Y mice, both sedentary and runners, spent a comparable time in the center of the arena (Figure 2D), indicating that the impulsive exploratory behavior of *Cdkl5* −/Y mice is not due to increased locomotor activity.

In order to examine the effect of voluntary running on motor stereotypies, mice were tested for hind-limb clasping (Figure 2E). Similar to sedentary *Cdkl5* −/Y mice, *Cdkl5* −/Y runners showed a high clasping time (Figure 2E), with a slight worsening trend, indicating that voluntary running does not have a positive impact on the stereotypic behavior that is due to loss of Cdkl5 expression.

To examine whether voluntary running improves cognitive impairment in *Cdkl5* −/Y mice, memory was evaluated using the passive avoidance (PA) test, a cognitive paradigm in which *Cdkl5* −/Y mice are documented to be impaired [8]. *Cdkl5* −/Y mice were severely impaired in this task, as shown by a reduced latency to enter the dark compartment the day after the experienced aversive event in comparison with wild-type (+/Y) mice (Figure 2F). In *Cdkl5* −/Y runners, the latency underwent an increase and was no different in comparison with that of *Cdkl5* +/Y mice (Figure 2F), suggesting memory improvement.

### 3.3. Effect of Voluntary Wheel Running on Hippocampal Neurogenesis and Neuronal Survival in Cdkl5 KO Mice

To establish the effect of voluntary running on hippocampal neurogenesis, we first evaluated the number of newborn cells present in the dentate gyrus (DG) using immunohistochemistry for doublecortin (DCX). DCX is a microtubule-associated phosphoprotein expressed by immature granule neurons during the period of neurite elongation, from one to four weeks after neuron birth, allowing an estimation of the number of new granule neurons. In agreement with previous evidence [12], *Cdkl5* −/Y mice had a reduced number of new granule cells in comparison with wild-type mice (Student *t*-test *p* < 0.0015, Figure 3A,B). In both wild-type and *Cdkl5* −/Y runners, there was an increase in the number of new neurons in comparison with their sedentary counterparts (Figure 3A,B): a 71 ± 14% and an 89 ± 20% increase, respectively.

To evaluate the effect of voluntary running on the survival of the newborn cells, the BrdU-positive cells present in the DG of mice injected with BrdU on the first day of running (one month before the sacrifice, Figure 1A) were counted. BrdU-positive cells were significantly increased in both the wild-type and the *Cdkl5* −/Y runner groups in comparison with their sedentary counterparts (Figure 3C; +/Y, 68 ± 16%; −/Y, 54 ± 21%), indicating an increased survival of newborn granule cells.

To establish the effect of increased neurogenesis on the net number of granule cells, we evaluated the granule cell density. The granule cell layer had a reduced cell density in *Cdkl5* −/Y compared to wild-type mice (Figure 3D). In both *Cdkl5* −/Y and wild-type mice, exercise significantly increased granule cell density (Figure 3D; +/Y, 4 ± 1.1%; −/Y, 11 ± 0.4%). These results indicate that voluntary running exerts a beneficial effect on hippocampal neurogenesis in both wild-type and *Cdkl5* KO conditions.

To further assess the effect of exercise on neuron survival, we evaluated the pyramidal neuron cell density in the CA1 layer of the hippocampus of sedentary and runner mice. As previously reported, *Cdkl5* −/Y mice [14,64] showed a low number of pyramidal neurons in the CA1 layer compared with their wild-type counterparts (Figure 3E). While in wild-type mice, voluntary running did not change the density of pyramidal neurons (Figure 3E), in *Cdkl5* −/Y mice, exercise restored the CA1 neuronal number to that of the wild-type condition (Figure 3E), suggesting that voluntary running exerts a generalized pro-survival effect on hippocampal neurons in *Cdkl5* −/Y mice.

### 3.4. Effect of Voluntary Wheel Running on Microglia Overactivation in Cdkl5 KO Mice

*Cdkl5* KO mice are characterized by increased microglial activation [14,15]. Since it has been demonstrated that exercise reduces the activation of microglia [65,66,67], we next explored whether voluntary running affects microglial cell status in the cortex and hippocampus of *Cdkl5* −/Y mice. The pro-inflammatory M1 state is characterized by an increased number of microglial cells, with larger and rounder cell bodies and thick protrusions. AIF-1 immunohistochemistry is extensively used to visualize changes in the number and morphology of microglial cells that are associated with this state. AIF-1-positive microglial cells had an enlarged body size and higher density in the *Cdkl5* −/Y cortex and hippocampus compared to their wild-type counterparts (Figure 4A–C). Wild-type runners showed no changes in microglia density and morphology compared to their sedentary counterparts (Figure 4A–C). Otherwise, voluntary running decreased microglia overactivation in *Cdkl5* −/Y mice, bringing it to the levels of the wild-type condition (Figure 4A–C), suggesting that exercise in *Cdkl5* −/Y mice exerts anti-inflammatory activity.

### 3.5. Effect of Voluntary Wheel Running on Spine Development in Cdkl5 KO Mice

Several studies have reported that loss of Cdkl5 negatively affects spine development and maturation [8,10,68,69]. Given the positive effects of voluntary running on neuronal survival and microglia activation in *Cdkl5* −/Y mice, we further investigated the effects of exercise on spine maturation. As previously reported [52], no difference in the spine density of Golgi-stained cortical pyramidal neurons was observed in sedentary *Cdkl5* −/Y mice (Figure 5A), nor does voluntary running have an impact on spinogenesis (Figure 5A).

Dendritic spines are heterogeneous in size and shape and can be classified as immature spines (filopodia, thin, and stubby-shaped) and mature spines (mushroom- and cup-shaped) (Figure 5B). Separate counts of different classes of dendritic spines revealed that the pyramidal neurons of *Cdkl5* −/Y mice had a higher percentage of immature spines (Figure 5C, both filopodia and thin) and a reduced percentage of mature spines compared to wild-type mice (Figure 5C; mushroom and cup). Voluntary running reduced the number of immature (filopodia and thin) spines and increased the number of mature (mushroom) spines (Figure 5C) in *Cdkl5* −/Y mice, restoring the balance between immature and mature spines in *Cdkl5* −/Y cortical pyramidal neurons (Figure 5D). A slight increase in dendritic spine maturation was also present in wild-type runners (Figure 5D).

### 3.6. Effect of Voluntary Wheel Running on BDNF Expression in Cdkl5 KO Mice

Next, we examined BDNF levels in the hippocampus of runner mice using immunohistochemistry and Western blot analysis. Confirming the results of previous studies [70,71], we found strong BDNF immunoreactivity in the mossy fibers and CA3 pyramidal cells (Figure 6A). As previously reported in female *Cdkl5* +/− mice [49], BDNF levels are not altered in the hippocampus of male *Cdkl5* −/Y mice (Figure 6B–D). As expected, voluntary running caused a significant increase in BDNF levels in wild-type mice (Figure 6B–D). A similar increase was observed in *Cdkl5* −/Y runners (Figure 6B–D).

## 4. Discussion

Exercise and voluntary wheel running have not only been shown to increase neurogenesis [28], but also to improve motor and cognitive tasks [72,73], exerting beneficial effects on diseases of the CNS, such as neurodevelopmental disorders [26,56,57,58]. To date, no studies have analyzed the effect of physical exercise on the phenotype of the *Cdkl5* KO mouse, a mouse model of CDD. Our study showed that long-term voluntary running improved the hyperlocomotion and impulsivity behaviors and memory performance of *Cdkl5* −/Y mice. This is correlated with (i) increased hippocampal neurogenesis and neuronal survival; (ii) inhibition of microglia activation; (iii) increased spine maturation, and (iv) BDNF levels.

In contrast to free walking, voluntary wheel running implies a series of complex movements and motor skill learning [74]. We found that wild-type and *Cdkl5* −/Y mice displayed comparable daily rhythms during the first week of running. However, after the first week, while wild-type mice had the ability to learn this kind of complex movement and progressively increase their total running distance, *Cdkl5* −/Y mice did not increase the distance covered, suggesting an impaired skilled motor function/learning. This is not surprising since *Cdkl5* KO mice are characterized by impaired skilled motor function of forepaws [75] and reduced motor coordination and learning in a forced locomotion task (rotarod) [6,9,64]. However, since we did not evaluate the time that the mice spent running on the wheel, we cannot exclude that lower motivation for this activity may contribute, at least in part, to the diminished running of the *Cdkl5* KO mouse. Recent evidence highlighted that Cdkl5 plays a role in normalizing synaptic dopamine tone and, consequently, movement control [61]. Accordingly, previous findings showed that the decline in motor skills in middle-aged *Cdkl5* KO mice correlates with decreased survival of tyrosine hydroxylase (TH)-positive neurons [64]. Since motor skill learning in a running wheel task is also dependent on dopaminergic inputs [74], it is possible that synaptic aberrant dopamine tone may play a role in the impaired running skills of *Cdkl5* KO mice.

Even though the *Cdkl5* −/Y mice ran much less vigorously than the wild-type mice, an evident behavioral improvement was present. In particular, hyperlocomotion and exploratory impulsivity in an open-field arena were reduced to the wild-type levels, suggesting a recovery of symptoms that are typical of attention-deficit hyperactivity disorder (ADHD). Importantly, the time spent in the center of the open-field arena did not differ between the groups, indicating that anxiety-like behavior did not bias the outcome of this motor test. Our study is the first to demonstrate that *Cdkl5* KO mice show an increased exploratory impulsivity in reaction to objects in an open-field arena, but this finding is in accordance with a recent finding showing that *Cdkl5* KO mice display an impulsive and hyperactive phenotype in an appetitive conditioning task [63]. We found a large inter-individual variability in the impulsive and hyperactive phenotype of *Cdkl5* KO mice. Although we do not have an explanation for the different behavioral attitude of *Cdkl5* −/Y mice, a large individual variability was also found in a behavioral impulsivity test based on an appetitive conditioning task in *Cdkl5* KO mice [63]. Our findings showed that voluntary running does not reduce the hind-limb clasping behavior of *Cdkl5* KO mice; rather, this motor deficit is aggravated. This is not surprising, since the effect of running on clasping behavior is less clear. For instance, while mice with Huntington’s disease that had access to running wheels exhibited delayed onset of hind-limb clasping [76], a worsening of the clasping behavior was described in a running mouse model of Alzheimer’s disease [77]. The causes of the aggravation of this pathological phenotype are unclear at the moment.

The Morris water maze is most often used to analyze learning and memory in rodents, including *Cdkl5* KO mouse models [8,9]; however, we did not use this test to analyze cognitive function, so as to avoid confounding results due to the swimming deficits of *Cdkl5* KO mice [9] and the visual problems recently disclosed in these mice [78]. Here, we used the passive avoidance task, which is a fear-motivated test used to assess hippocampus-dependent learning and memory in rodents [79]. The results suggest that voluntary wheel running improves memory in *Cdkl5* −/Y mice. However, due to the large inter-subject variability in fear memory retention in both wild-type and *Cdkl5* −/Y mice, which could depend on a variable shock sensitivity and/or anxiety state, the number of *Cdkl5* −/Y runners used in this study was not sufficient to reach statistical significance. As it is widely assumed that the beneficial effects of voluntary exercise on cognition may be caused by increased hippocampal neurogenesis, a process that decreases with the progression of aging, we can speculate that wheel running from a juvenile age might be more effective in rescuing the cognitive deficit in *Cdkl5* KO mice.

The effects of voluntary running on survival of newborn DG cells are well documented [80]. Here, we exposed BrdU-injected adult mice to running wheels for a month and demonstrated that running mice (both wild-type and *Cdkl5* −/Y mice) have increased BrdU-positive cells in the hippocampal DG. Since the BrdU-positive cells were analyzed at 4 weeks post-BrdU injection, this increase in BrdU-labeled cells is likely a result of both increased precursor proliferation and newborn neuron survival in the DG. We found that voluntary running also prevents cell death of the CA1 hippocampal neurons caused by the loss of *Cdkl5* [17,64]. Several lines of evidence indicate that exercise modulates multiple systems that are known to regulate neuroinflammation [66] and prevents neuronal death of the hippocampal neurons resulting from an inflammatory response [81]. Activated microglia play active roles in the pathogenesis of CDD, as demonstrated by the therapeutic efficacy of luteolin, a natural anti-inflammatory flavonoid, in restoring hippocampal neuronal survival in *Cdkl5* KO mice [14]. Our finding that voluntary running inhibits microglia overactivation in the brain of *Cdkl5* KO mice could underlie the restoration of hippocampal neuron survival. However, the exercise-related decreased microglial activation is generally considered an indirect consequence of reduced neuronal injury elicited by the BDNF neurotrophic effect [66].

BDNF is one of the most-studied neurotrophic factors whose production in the brain is increased by exercise [82,83,84]. BDNF can regulate brain functions in multiple aspects, including neuronal cell survival, adult hippocampal neurogenesis, and neuroplasticity [85]. Furthermore, BDNF can alleviate microglial activation in several brain disease models [86,87,88,89]. Therefore, we can strongly hypothesize that increased levels of BDNF underlie exercise-induced structural and behavioral improvements in *Cdkl5* KO mice. Dendritic spines are another morphological feature of neurons that voluntary running has been shown to affect [90]. The exercise-dependent stimulation of spine maturation in newborn neurons has been demonstrated by several studies [35,91,92,93,94]. Here, we report that voluntary running increases spine development of cortical pyramidal neurons in *Cdkl5* KO mice, spurring a conversion of dendritic filopodia to mature spines. The physiological relevance of the positive effects of BDNF signaling on dendritic spine formation/maturation is supported by several observations [95]. Loss- and gain-of-function experiments of BDNF in vitro affect dendritic spine morphology, thereby altering the distribution of different spine types. In vivo, mutant mice that specifically lack the dendritic localization of BDNF show a higher density of longer and thinner dendritic spines in cortical pyramidal neurons and in the hippocampus [96]. In line with this evidence, we recently demonstrated that the memory impairment as well as the spine maturation of perirhinal cortical neurons is rescued by a treatment with a TrkB agonist in *Cdkl5* KO mice (the 7,8-DHF prodrug R13 [55]). Moreover, pharmacological enhancement of 5-HT neurotransmission using sertraline increases BDNF expression levels and restores dendritic spine maturation in the brain of *Cdkl5* +/− mice, strengthening the link between exercise-related BDNF boosting and spine recovery.

The finding that voluntary running has relatively scarce advantages in wild-type mice is in line with similar evidence in control runners [97] and suggests that exercise, and consequently BDNF signaling activation, has greater ameliorative effects in abnormal than in normal brain conditions.

## 5. Conclusions

Our study demonstrates that *Cdkl5* KO mice benefit from sustained voluntary running, suggesting that physical exercise may be beneficial in individuals with CDD, as it is in healthy humans. We are aware that voluntary exercise may not be feasible for some CDD patients; however, since it has been shown that involuntary or forced exercise, similarly to voluntary exercise, can improve cognitive impairment as well as induce BDNF expression [98], our findings provide support for the idea that a properly designed physical exercise program could be a valuable adjuvant to a future pharmacotherapy for CDD.

## Figures and Tables

**Figure 2 biomolecules-13-01396-f002:**
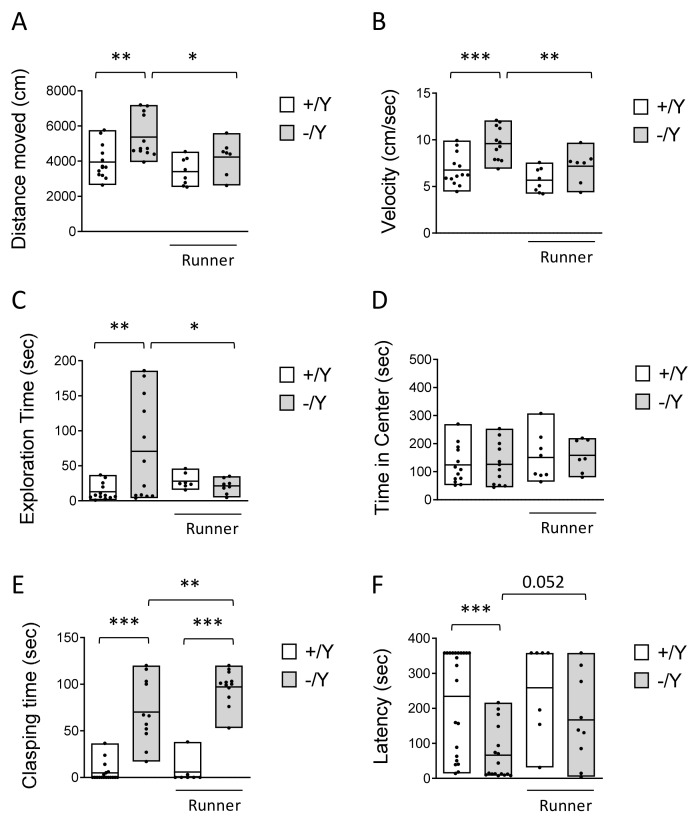
Effect of voluntary wheel running on behavior in *Cdkl5* −/Y mice. (**A**,**B**) Locomotor activity measured as the total distance travelled (**A**) and average locomotion velocity (**B**) during a 10 min open-field test in sedentary *Cdkl5* +/Y (*n* = 13) and *Cdkl5* −/Y (*n* = 12) mice and in *Cdkl5* +/Y (*n* = 8) and *Cdkl5* −/Y (*n* = 7) mice after the one-month period of voluntary wheel running exercise (Runner). (**C**,**D**) Time (cumulative duration) spent by mice as in B exploring the two objects located in the open-field arena (**C**) or moving in the center of the open-field arena (**D**). (**E**) Total amount of time spent hind-limb clasping during a 2 min interval in *Cdkl5* +/Y (*n* = 18) and *Cdkl5* −/Y (*n* = 11) mice and in voluntary wheel running *Cdkl5* +/Y (*n* = 7) and *Cdkl5* −/Y (*n* = 12) mice. (**F**) Passive avoidance test in *Cdkl5* +/Y (*n* = 22) and *Cdkl5* −/Y (*n* = 17) control mice and in *Cdkl5* +/Y (*n* = 7) and *Cdkl5* −/Y (*n* = 9) runner mice. Graphs show the latency to enter the dark compartment on the 2nd day of the behavioral procedure. Bar plots represent min, max, and mean values. * *p* < 0.05, ** *p* < 0.01, *** *p* < 0.001, Fisher’s LSD test after two-way ANOVA.

**Figure 3 biomolecules-13-01396-f003:**
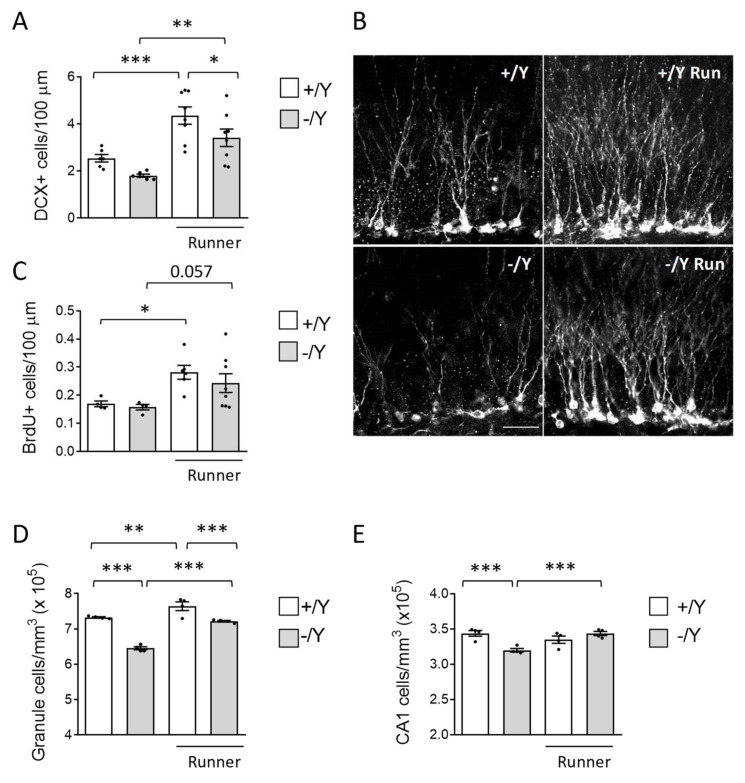
Effect of voluntary wheel running on hippocampal neurogenesis of *Cdkl5* −/Y mice. (**A**) Quantification of DCX-positive cells in the granular layer (GL) of the dentate gyrus (DG) of sedentary *Cdkl5* +/Y (*n* = 6) and *Cdkl5* −/Y (*n* = 6) mice and of *Cdkl5* +/Y (*n* = 8) and *Cdkl5* −/Y (*n* = 8) mice after one month of voluntary wheel running. (**B**) Representative image of a section of the upper GL from the DG of one animal per experimental group processed for DCX immunostaining. Scale bar = 40 μm. (**C**) Number of BrdU-positive cells in the subgranular zone (SGZ) and GL of the DG of sedentary *Cdkl5* +/Y (*n* = 4) and *Cdkl5* −/Y (*n* = 4) mice and in *Cdkl5* +/Y (*n* = 6) and *Cdkl5* −/Y (*n* = 8) runner mice. (**D**,**E**) Quantification of Hoechst-positive cells in the upper GL of the DG (**D**) and in the CA1 field of the hippocampus (**E**) of sedentary *Cdkl5* +/Y (*n* = 4) and *Cdkl5* −/Y (*n* = 4) mice and of *Cdkl5* +/Y (*n* = 4) and *Cdkl5* −/Y (*n* = 4) runner mice. Data in A and C are expressed as number of DCX- or BrdU positive cells per 100 µm. Data in (**D**,**E**) are expressed as number of Hoechst-positive cells per mm^3^. Values in (**A**,**C**–**E**) are represented as means ± SEM. * *p* < 0.05, ** *p* < 0.01, *** *p* < 0.001, Fisher’s LSD test after two-way ANOVA.

**Figure 4 biomolecules-13-01396-f004:**
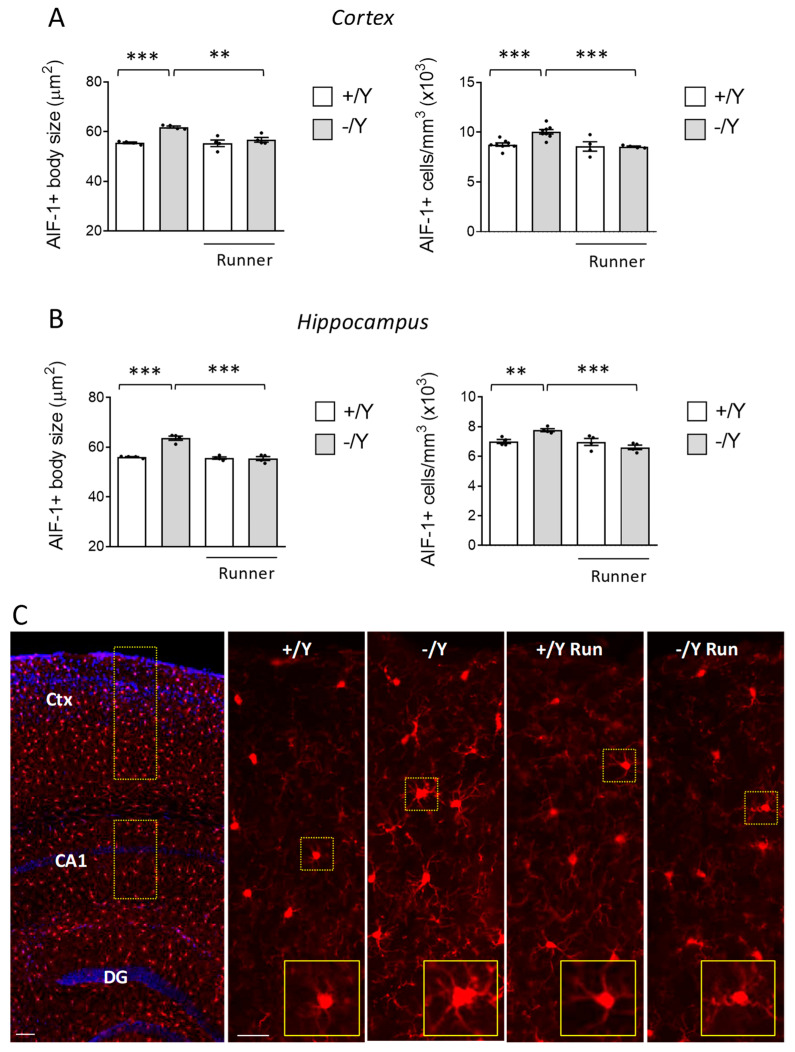
Effect of voluntary wheel running on neuroinflammation in *Cdkl5* −/Y mice brain. (**A**,**B**) Mean microglia cell body size (histograms on the left) and quantification of AIF-1-positive cells (histograms on the right) in the somatosensory cortex (**A**) and in the hippocampus (**B**) of sedentary *Cdkl5* +/Y (*n* = 4) and *Cdkl5* −/Y (*n* = 4) mice and of *Cdkl5* +/Y (*n* = 4) and *Cdkl5* −/Y (*n* = 4) mice after one month of voluntary wheel running. (**C**) Representative image of an AIF-1-stained section (panel on the left; scale bar = 100 μm), showing the portion of cortical and hippocampal regions in which microglial AIF-1-positive cells were evaluated (areas enclosed by the dashed rectangles). On the right are examples of representative fluorescence images of cortical sections processed for AIF-1 immunohistochemistry of one mouse for each experimental condition. The dotted boxes in the upper panels indicate microglial cells shown in magnification in the lower panels. High magnification (scale bar = 10 μm) and low magnification (scale bar = 25 μm) are shown. Values in (**A**,**B**) are represented as means ± SEM. ** *p* < 0.01, *** *p* < 0.001. Fisher’s LSD test after two-way ANOVA. Abbreviations: Ctx = cortex, CA1 = hippocampal CA1 field, DG = dentate gyrus.

**Figure 5 biomolecules-13-01396-f005:**
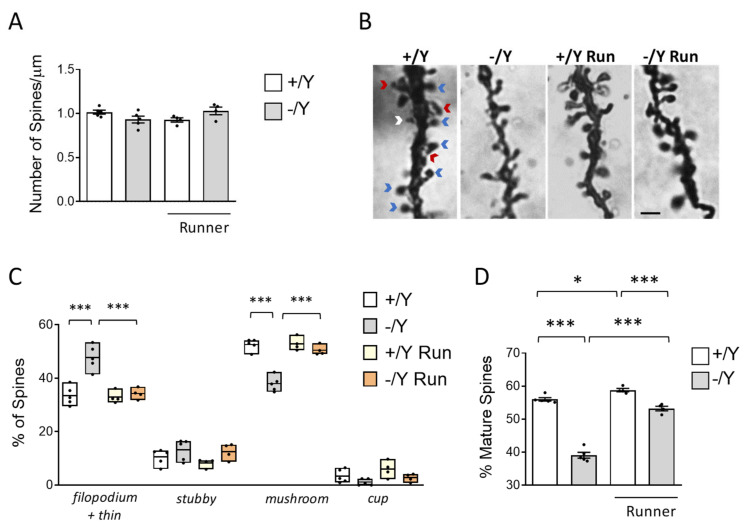
Effect of voluntary wheel running on cortical dendritic spine maturation in *Cdkl5* −/Y mice. (**A**) Comparison of spine density in basal dendrites of cortical pyramidal neurons of sedentary *Cdkl5* +/Y (*n* = 4) and *Cdkl5* −/Y (*n* = 4) mice and of *Cdkl5* +/Y (*n* = 4) and *Cdkl5* −/Y (*n* = 4) mice after one month of voluntary wheel running. (**B**) Examples of Golgi-stained cortical pyramidal dendrites of one animal from each experimental group. Red arrows represent immature filopodium-like or thin spines, the white arrow represents an immature stubby-shaped spine, and blue arrows represent mature mushroom-shaped spines. Scale bar = 2 μm. (**C**) Percentage of immature and mature dendritic spines of each morphological class in relation to the total number of protrusions in cortical pyramidal neurons of *Cdkl5* +/Y and *Cdkl5* −/Y mice as in A. (**D**) Percentage of mature dendritic spines over the total spine number in basal dendrites of cortical pyramidal neurons of *Cdkl5* +/Y and *Cdkl5* −/Y mice as in A. Values are represented as means ± SEM. * *p* < 0.05, *** *p* < 0.001. Fisher’s LSD test after two-way ANOVA.

**Figure 6 biomolecules-13-01396-f006:**
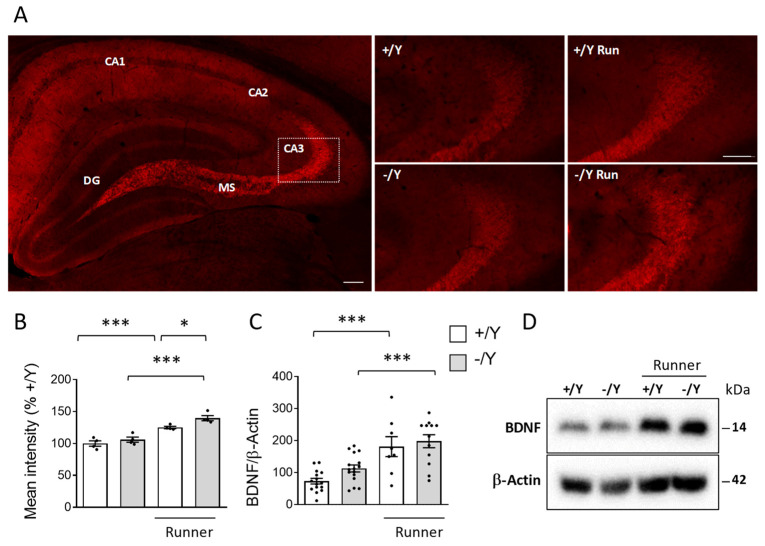
Effect of voluntary wheel running on BDNF expression in the hippocampus of *Cdkl5* −/Y mice. (**A**) Representative image of a BDNF-stained section, showing the portion of the hippocampal region in which the intensity of BDNF staining in presynaptic terminals of hippocampal mossy fibers and CA3 pyramidal cells was quantified. The white dotted box in the panel indicates the position of the high magnifications of the CA3 layers of one mouse for each experimental condition. High magnification (scale bar = 100 μm) and low magnification (scale bar = 100 μm) are shown. (**B**) Quantification of the mean intensity of BDNF immunoreactivity per area in mossy fiber terminals of the hippocampal region of sedentary *Cdkl5* +/Y (*n* = 4) and *Cdkl5* −/Y (*n* = 4) mice and of *Cdkl5* +/Y (*n* = 4) and *Cdkl5* −/Y (*n* = 4) mice after one month of voluntary wheel running. Data are given as percentages in relation to sedentary *Cdkl5* +/Y mice. (**C**,**D**) Western blot analysis of BDNF and β-Actin levels in hippocampal homogenates from sedentary *Cdkl5* +/Y (*n* = 15) and *Cdkl5* −/Y (*n* = 16) mice and from *Cdkl5* +/Y (*n* = 8) and *Cdkl5* −/Y (*n* = 12) mice after one month of voluntary wheel running. The histogram in (**C**) shows mature BDNF protein levels normalized to β-Actin. Data are expressed as percentages in relation to *Cdkl5* +/Y mice. Examples of immunoblots in (**D**). Values are represented as means ± SEM. * *p* < 0.05, *** *p* < 0.001 (Fisher’s LSD test after two-way ANOVA). Abbreviations: CA1 = hippocampal CA1 field, CA2 = hippocampal CA2 field, CA3 = hippocampal CA3 field, DG = dentate gyrus, MS = mossy fibers.

## Data Availability

Data will be made available on request.

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
