# Peer review of "Voluntary Running Improves Behavioral and Structural Abnormalities in a Mouse Model of CDKL5 Deficiency Disorder"

_biomolecules, 2023, doi:10.3390/biom13091396_

Round 1

Reviewer 1 Report

Introduction:

The authors state on line 51 “…and an increased status of microglia activation” and on line 66 “…and stimulating the expression of a pro-neurogenic microglial phenotype”. What kind of microglial activation is observed? What markers are quantified? M1 or M2 phenotype?

Very simple and direct introduction with the necessary information to understand the aim of the work.

Materials and Methods:

Line 101 “5-bromo-2′-deoxyuridine (BrdU,)” and line 123 “Bromodeoxyuridine (BrdU)”, Refer only once and standardize.

Line 101 “5-bromo-2′-deoxyuridine (BrdU; 150 mg/kg, Sigma, USA”, Please provide the catalog number.

Please provide the catalog number of all reagents (e.g.  BrdU; Rapid Golgi Stain TM Kit. DPX mounting medium…) and kits used as well as of all equipment’s (e.g. cryostat, microtome…).

Line 103: “Running in the home cage”. My first question is how did the authors record the path/number of revolution that each of the animals in the cage did individually? This was not clear to me.

My second question is related to the fact that in each of the cages there could be between 3 and 5 animals (as previously mentioned). The more animals the greater the path/number of revolutions they could reach throughout the day, taking this into account did the authors do any normalization taking into account the number of animals that were in the cage and the number of path/revolutions? Or the electronic device have an individualized record of each animal?

I think the work would gain more quality if a subsection was added at the beginning of the materials and methods section to explain the experimental procedure. It would allow the reader to have an overview of the work before reading in detail each of the techniques used.

Results:

            Line 241: Why Cdkl5 +/Y have an n=8 and Cdkl5 -/Y an n=12?

            Line 267: In the graph 2C, N=7 of the -/Y animals (out of 13) behave similarly to the +/Y control animals, what does this mean? This should be discussed.

Line 270: Why the difference in the number of elements in each experimental condition? Why does it change from 13 to 7 (“sedentary Cdkl5 +/Y (n = 13) and Cdkl5 -/Y (n = 11) mice and in Cdkl5 +/Y (n = 8) and Cdkl5 -/Y (n = 7)”? Or for example “Passive avoidance test in Cdkl5 +/Y (n = 22) and Cdkl5 -/Y (n = 17) control mice and in Cdkl5 +/Y (n = 7) and Cdkl5 -/Y (n = 9)”?

            Line 288: On figure 2F, there are at least 7 animals on +/Y sedentary condition (out of 22) that behave exactly as -/Y sedentary condition, what this means?

            About figure 2, it seems to me that some of the considerations, especially those associated with graphs 2C and 2F seem to me to be far-fetched and require explanation.

            Line 339: What kind of microglial marker is AIF-1? It is associated with what type of microglial phenotype? In other words, its expression is associated to an pro-inflammatory microglial phenotype or is associated to an beneficial microglial phenotype? Thys should be referred in the manuscript.

            Line 413: “Cdkl5 +/Y (n = 15) and Cdkl5 -/Y (n = 16) mice and from Cdkl5 +/Y (n = 9) and Cdkl5 -/Y (n = 15)”. Why the difference in the number of animal between Cdkl5 +/Y and the other experimental conditions?

            It is a very interesting paper on a relatively new topic. The introduction seems to be well constructed and with enough information on the topic to understand the work. The materials and methods must be completed with more information as mentioned above. Regarding the results, most of the data seems to be well supported, however I think it is necessary for the authors to clarify some of the doubts that they left me when evaluating the work.  

Reviewer 2 Report

In general, I think the subject of this article is really interesting, and the authors’ fascinating observations on this timely topic may be of interest to the readers of Biomolecules. However, some comments needed to be addressed to improve the quality of the manuscript prior to its publication in the present form. My overall judgment is to publish this paper after the authors have carefully considered my suggestions below.

Majors:

1) The data on the mean number of revolutions for every week and body weight should be re-analyzed with ANOVA test for repeated measures. The changes in body weight of wheel revolutions measured in the same animals at different time points are a dynamic parameter and the analysis of these data as independent variables is inappropriate.

2) The adequate description of statistical data is absent throughout the “results” section, t-test or Fisher’s test values are not indicated as well as degrees of freedom.

3) Why does the number of animals vary from one behavioral test to another? For example, the legend for Figure 1 indicates that the number of sedentary Cdkl5 +/Y mice is 15, but in passive avoidance test was used 22 control (i.e. sedentary) Cdkl5 +/Y mice.

4) The authors indicated that “…an operator annotated the number of rotations performed by each mouse…” (lines 108-109). But it is not clear from the description how the running was tracked for each animal given that all animals housed in groups.

Minor:

The sentences from abstract (lines 13-15) and introduction (lines 39-41) duplicate each other.

Round 2

Reviewer 2 Report

I am completely satisfied with the answers of the authors. I have no more comments.